# Effects of Message Framing and Information Source on Consumers’ Attitudes toward an Amino Acid-Based Alternative Meat Curing System

**DOI:** 10.3390/foods12071535

**Published:** 2023-04-05

**Authors:** Amber Vonona Chambers, Mathew T. Baker, Holli R. Leggette, Wesley N. Osburn, Peng Lu

**Affiliations:** 1Department of Agricultural Leadership, Education and Communications, Texas A&M University, College Station, TX 77843, USA; 2Department of Animal Science, Texas A&M University, College Station, TX 77843, USA

**Keywords:** message framing, information source, consumer attitudes, novel meat curing system

## Abstract

Recently, meat scientists have developed an innovative amino acid-based alternative meat curing system (AAACS). However, consumer skepticism toward novel foods presents challenges regarding the acceptance of food innovations like the AAACS. Effective communication about this and other food technologies is critical. Our study was a 2 × 4 randomized factorial between-groups experiment that investigated how two peripheral cues—message frame and information source—impact attitudes toward the AAACS. We used Qualtrics to randomly assign participants to one of eight treatment groups. Each group viewed a different video about the AAACS. Then, all participants were asked about their attitudes toward the alternative meat curing system. Data were analyzed using a two-way multivariate analysis of variance (MANOVA). The two-way MANOVA determined concurrently the experimental effects of message frame and information source on information recall, trust, source expertise, source credibility, and anticipated consumption behavior. A significant MANOVA was followed up using Discriminant Function Analysis (DFA). A significant main effect was found for information source. The DFA revealed only one significant underlying function and that source expertise was the most powerful discriminating variable for information source.

## 1. Introduction

Currently, most meat products are cured through the direct addition of sodium nitrite, which is a reactive crystalline salt that can function as a reducing, oxidizing, or nitrosylating agent. Sodium nitrite can be converted into many different compounds, including nitric oxide [1]. Nitrite is an important additive for cured meat products because it contributes to cured meat color and flavor, aids in the suppression of microbial growth, and inhibits oxidation. Despite the benefits of sodium nitrite, consumers have expressed concerns regarding the consumption of cured meat products containing sodium nitrite [2,3]. For example, in the 1970s, concerns regarding nitroso compounds potentially yielding carcinogenic nitrosamines resulted in the near-ban of products cured with sodium nitrite [4]. Since then, other concerns related to cancer and leukemia have been directly related to meat products cured by the direct addition of sodium nitrite [3].

Therefore, to address consumer concerns, meat-curing alternatives such as tomato paste [5], herbs and berries [6], and vegetable powder [7] have been developed. Yet, many of these alternatives have undesirable organoleptic properties including a less intense cured meat color and a vegetable taste/aroma. Currently, there is no single ingredient that can replace the functionality (color, flavor, shelf life, and safety) of curing meat with sodium nitrite.

Given the demand for alternative meat curing methods, meat scientists have developed an innovative alternative meat curing system. The description material of the meat curing system has been included as Appendix A. This system replaces the direct addition of sodium nitrite with the naturally occurring amino acid L-arginine. L-arginine activates the endothelial nitric oxide synthase (eNOS) system found in muscle/meat, which produces nitric oxide and another amino acid, L-citrulline. The eNOS system is vital for muscle function because of its ability to convert L-arginine to nitric oxide [7]. The eNOS system forms nitric oxide and oxygen simultaneously. Nitric oxide improves vasodilation and muscle metabolism [8]. Nitric oxide is oxidized to nitrite and then to nitrate in the presence of oxygen [9]. Nitric oxide produces the cured meat color via binding of the nitric oxide to the meat pigment, myoglobin, and develops the traditional cured pink color during heating. Two nitric oxide molecules can form residual nitrite, which contributes to antimicrobial and antioxidant properties that improve shelf life and safety [7]. Fundamentally, this alternative amino acid-based curing system is a novel way to cure meat without synthetic sodium nitrite.

In the last three decades, the food industry has become consumer-driven, and consumers’ attitudes affect the structure and management of the food system [10]. For example, when introducing innovations like the AAACS, consumer skepticism toward novel foods and food technologies presents challenges and must be addressed before it can become a widely adopted technology [11]. Because of this, it is critical that the first step in communicating about the alternative curing system is to focus on consumers’ knowledge, awareness, and understanding. Bridging the information gap is critical to the acceptance of the alternative curing system and to shaping positive attitudes about the AAACS. Thus, information regarding the AAACS needs to be communicated strategically and effectively. However, doing so is complex. Wide gaps have been found between scientific and public opinion regarding many controversial scientific topics including climate change, genetically modified food, and chemicals used in food production [12]. Effective communication, grounded in theory and refined through empirical testing and assessment, can address this divide.

Two theoretical frameworks—the elaboration likelihood model (ELM) and message framing—guided our study. According to the elaboration likelihood model, information is processed through either a central or peripheral processing route [13]. The ELM refers to the extent someone will consider a persuasive message [14,15]. Individuals process unfamiliar information passively through the peripheral route based on message-specific peripheral cues [14]. Peripheral cues include message characteristics like the number of arguments, message frame, source features, graphic elements, credibility, etc. After information is processed through the peripheral route, individuals may seek further information to be processed actively through the central processing route [13]. Because most consumers are not familiar with the AAACS, the study described herein investigated the experimental effect of two peripheral cues—message frame and information source—on a combined set of dependent variables including information recall, trust, source expertise, source credibility, and anticipated consumption behavior of meat products cured with the AAACS.

One important peripheral cue that affects how information is processed in the peripheral processing route is the way a message is framed. Message framing is a communication strategy that determines and limits what is communicated from the actual events and contributes to organizing and making sense of the information [16]. Framing can determine what people notice about the message, how they comprehend the information, how they remember the information, and how they react to the message being communicated [17]. In the context of science communication, message framing is used to emphasize or highlight relevant aspects of scientific findings [18]. Narrative message framing is one strategy used to provide specific cases that require generalizing up to a general truth [19]. Narrative framing uses personal stories, anecdotes, perspectives, and appeals to support the facts around an issue. Narrative framing puts more emphasis on the engaging, interesting, and easily comprehended aspects of information [20]. Narrative and analytical message frames are often contrasted in framing research [21]. Analytical communication attempts to provide abstract truths that can remain true through various situations with individuals generalizing down to a specific case [19]. Analytical information follows deductive reasoning while narrative information follows inductive reasoning [20].

Individuals’ perceptions of the messenger, or source of information, is another important peripheral cue that can influence the passive processing of information [22]. Previous research indicates that source credibility and source expertise affect outcomes including attitudes, disposition towards information, and intention to consume [23,24,25,26,27]. Credible sources are more persuasive than less-credible sources [28], and highly credible sources induce more behavioral compliance than less-credible sources [29,30]. Messages delivered by a more credible individual may have a more meaningful impact than a message delivered by someone who is perceived as less credible [31]. When manipulating source expertise, sources perceived as experts generate opinion change. Furthermore, trustworthy communicators generate the most opinion change, regardless of whether they are perceived as an expert or not [32]. It is important to consider the source of information when exploring the effectiveness of various communication strategies. The acceptance of scientific information not only depends on the way information is presented but who is presenting the information [33].

We used a randomized controlled experimental design to determine how message frame and/or the source of information impact participants’ information recall, trust, source expertise, source credibility, and anticipated consumption behavior of products cured using the alternative meat curing system. We assigned participants to watch one of eight videos that simulated real-world communication conditions as closely as possible. We examined the impact using three research questions: (1) Is there a significant interaction between message frame and information source on combined dependent variables of information recall, trust, source expertise, source credibility, and anticipated consumption behavior? (2) Are there significant differences in the combined dependent variables of information recall, trust, source expertise, source credibility, and anticipated consumption behavior for message frame (analytical/narrative)? (3) Are there significant differences in the combined dependent variables of information recall, trust, source expertise, source credibility, and anticipated consumption behavior for information source (consumer/producer/reporter/meat scientist)?

## 2. Materials and Methods

We used a 2 × 4 randomized factorial between-groups experimental design and developed an online, 22-item instrument using Qualtrics. Four rounds of recruitment emails with a link to the instrument were sent through a university email system between March and April 2022. The survey included a screening question to ensure participants represented only students who were part of Generation Z. Generation Z was chosen as the target population because this generation includes current and primary shoppers and emerging primary shoppers. Their unique preferences have an important impact on the food industry [34]. A non-random sample was taken of the population. The sample included students at a large land-grant university in the southern United States. We programmed Qualtrics to assign all participants randomly to view one of eight versions of a video about the AAACS. We used videos to disseminate variations of the message because they are one of the most powerful storytelling mediums that can be used to promote products [35]. Each of the eight videos was approximately two minutes long because previous research indicates that videos between zero and three minutes have the highest engagement [36].

### 2.1. Participants

A total of 470 participants accessed the instrument. Incomplete questionnaires were deleted from the dataset. After cleaning the data, the sample size was 266 (n = 266). In terms of gender, 55.64% of the students indicated they were female (n = 148), 31.2% (n = 83) male, 1.5% (n = 4) nonbinary, and 11.6% (n = 31) preferred not to answer. More than half of the participants indicated they were white (n = 168, 63.16%) with 11.28% (n = 30) indicating they were Hispanic, Latino, or Spanish origin. Additionally, 6.02% of students indicated they were part of multiple ethnicity categories, and multiple ethnicities were represented by the sample. Most respondents were undergraduate students (n = 177, 66.54%), with 20.34% of students indicating they were graduate students (20.30%). Fourteen colleges, schools, or programs within the university were represented. The largest group of students was part of the College of Agriculture and Life Sciences (n = 73, 27.44%), followed by the College of Architecture (n = 30, 11.28%), the College of Liberal Arts (n = 27, 10.15%), and the College of Engineering (n = 24, 9.02%). Finally, participants were asked about their cured meat consumption. The largest group of students indicated consuming cured meat a few times a week (n = 69, 25.94%). The next largest group indicated consuming cured meat a few times a month (n = 55, 20.68%), followed by once a week (n = 30, 11.28%), and monthly (n = 21, 7.90%).

### 2.2. Research Variables

#### 2.2.1. Independent Variables

The first independent variable we tested was message frame. We developed two video scripts. One script was narratively framed, and one script was analytically framed. The video script used to create the videos operationalized message frame and differentiated the narrative message frame from the analytical message frame. We developed and transcribed message scripts for each message from actual events and presented them to the participants in video format. The narrative video script connected to participants’ personal lives and appealed to the general food preferences of Generation Z, including the social component of food consumption [37] and sustainability [38]. Generation Z stands out for placing value on social relations or settings (barbeques, tailgates, family dinners, etc.) as a motive for buying food [37]. In addition, sustainability has started permeating the lifestyle and food consumption of Generation Z [38]. In the narrative message script, cured meat products were discussed in the context of social settings such as barbeques and holiday dinners. Additionally, the narrative message alluded to the idea that the innovative cured meat system was one way to continue to produce cured meat products as a more sustainable food choice. A more personal tone, less formal language, and pronouns were included in the narrative video script. Conversely, the analytical video script used deductive reasoning [20] in which individuals generalized down to a specific case [20]. The analytical message frame relied heavily on objective scientific evidence including results from previous experimental research trials to support claims about the AAACS. The analytical message did not use pronouns, included technical language used by meat scientists, and somewhat resembled a scientific abstract [19,33]. When developing the analytical message script, we did not attempt to connect to the personal lives of individuals. The analytical video script was context-free and communicated an abstract truth that can remain true in a variety of situations [19]. The narrative and analytical message scripts have been included as Appendix A. Message frame was measured at the nominal level (1 = analytical; 2 = narrative).

The second independent variable we tested was information source. The message was delivered by a consumer, producer, reporter, or meat scientist. The eight videos included a consumer delivering the analytical message using the analytical video script, a consumer delivering the narrative message using the narrative video script, a producer delivering the analytical message, a producer delivering the narrative message, a reporter delivering the analytical message, a reporter delivering the narrative message, a meat scientist delivering the analytical message, and a meat scientist delivering the narrative message. We held other video elements and message characteristics constant, including length, titles, actor, and enthusiasm. We uploaded all videos to a personal YouTube account for closed captioning. Participants watched one of eight versions of the video, then were asked to recall information about the alternative meat curing system. Next, participants were asked to indicate their trust toward the message, report their perception of source expertise and credibility, and select their anticipated future consumption of products cured with the AAACS. Demographic data were collected at the end of the instrument.

#### 2.2.2. Dependent Variables

This study examined participants’ attitudes toward the AAACS after watching one of eight videos about the alternative curing system. Together, the five dependent variables represented a measure of all three components of an attitude. The tripartite classification of attitudes breaks attitudes into three components—cognitive, affective, and behavioral [39]. Information recall was a measure of the cognitive component of an attitude. Trust, source expertise, and source credibility were measures of the affective component of an attitude. Finally, anticipated consumption behavior was a measure of the behavioral component of an attitude.

Information recall evaluated how well participants were able to recall relevant information about the AAACS that was presented in the video they viewed. A meat scientist developed four multiple-choice questions based on what knowledge of the AAACS he hopes a potential consumer would possess after being exposed to marketing or communication material. Information recall was chosen as a cognitive measure of an attitude and designed this way to provide data most relevant and useful to a real-world application. Participants received a score from 0–4 to measure the dependent variable information recall, with each correct answer earning one point.

After an extensive instrument development process, three dependent variables, trust, source expertise, and source credibility were selected as affective measures of an attitude. In the initial design, only two dependent variables were selected as affective measures of an attitude. Trust was first selected due to the context of the innovation. Additionally, because information source was selected as an independent variable, source credibility was chosen as the other affective measure of an attitude.

The five items selected to measure trust came from Ohanian’s 1990 instrument [28]. This instrument was meticulously developed using psychometric scale-development procedures and was rigorously tested [28]. This instrument’s use of bipolar adjectives fit very well with the intended design of our instrument. Additionally, Ohanian’s instrument has been applied to various contexts including an emerging economy [40], online, non-profit organization communication [41], online word-of-mouth in food blogs [42], and video media about agriculturalists [43]. The five sets of bipolar adjectives were measured on semantic differential scales. Each set of adjectives described a dimension of trust [28]. The adjectives presented on semantic differential scales were trustworthy/untrustworthy, reliable/unreliable, dependable/undependable, honest/dishonest, and sincere/insincere [28].

In the pilot instrument, source credibility was the only other dependent variable included as an effective measure of attitude. Five sets of bipolar adjectives or phrases came from Ohanian’s 1990 instrument [28] including expert/not an expert, experienced/inexperienced, qualified/not qualified, knowledgeable/unknowledgeable, and skilled/unskilled. After reviewing the literature about source credibility, we chose to include four items from Besley et al.’s 2021 instrument measuring public perceptions of scientists [44] along with the source credibility measures from Ohanian’s instrument.

We chose to supplement Ohanian’s source credibility items with Besley et al.’s instrument because Besley’s instrument was developed through the examination of existing source credibility scales and a national survey. The top journals in science, environmental, and risk communication were examined for all scales measuring trust, credibility, and fairness in the last five years. Highly cited scales from other disciplines were also included. Besley provided an extensive instrument that combined relevant trust, credibility, and expertise scales into one. The adjectives or phrases included from Besley’s instrument were competent/incompetent, has integrity/does not have integrity, has goodwill/does not have goodwill, and open/not open [44].

After the nine sets of adjectives were combined, an exploratory factor analysis was conducted to establish the uni-dimensionality of the factors. It was anticipated that the items would load onto two factors, trust and source credibility. However, the exploratory factor analysis resulted in three factors. Because of this, what we initially thought of as source credibility was divided into two measured variables—source credibility and source expertise.

Source expertise has been described as the extent to which a communicator is perceived to be a source of valid assertations [45]. We measured source expertise with six sets of bipolar adjectives on a semantic differential scale (1 = negative; 7 = positive). The adjectives or phrases presented on the semantic differential scales included expert/not an expert, experienced/inexperienced, knowledgeable/unknowledgeable, qualified/unqualified, skilled/unskilled, and competent/incompetent. We derived the first five items from Ohanian’s 1990 instrument and the final item from Besley et al.’s 2021 instrument. We measured source credibility with three items from Besley et al.’s instrument, which included integrity, goodwill, and openness. Source credibility was measured using three sets of bipolar adjectives or phrases measured on a seven-point semantic differential scale (1 = negative, 7 = positive) [41].

Using Mplus, we conducted a confirmatory factor analysis to ensure the construct validity of the scale of trust, source credibility, and source expertise (Table 1). The overall model fit was good with a Root Mean Square Error of Approximation (RMSEA) of 0.05, meeting the goodness fit criteria of less than 0.05. The Comparative-Fit Index (CFI) was 0.97 and the Tucker-Lewis Index (TLI) was 0.96, thus, both indices were larger than the goodness-fit criteria of 0.95. The chi-squared statistic was significant χ^2^ (91) = 1839.30, *p* < 0.001. However, the chi-squared was influenced by sample size. Thus, we cannot rely on the chi-squared test alone to assess the model fit if the sample size is greater than 200 [46].

To measure a participant’s anticipated consumption behavior of products cured with the AAACS, we used a six-point Likert scale of 1 = strongly disagree, 2 = disagree, 3 = somewhat disagree, 4 = somewhat agree, 5 = agree, 6 = strongly agree [47]. Because anticipated consumption behavior was chosen as a measure of the behavioral component of attitude, no midpoint was included in the Likert scale. We decided to have participants select one side of the scale, to avoid the midpoint yielding potentially incorrect data. Participants can use the midpoint of a Likert scale as a dumping ground for a variety of reasons. To accurately measure the behavioral component of attitude, it was important to ask participants to choose which side of the measure they fell on. Four consumption-oriented questions were asked regarding various levels of consumption behavior participants anticipated.

### 2.3. Data Analysis

We analyzed data using SPSS v.28. We used two-way multivariate analysis of variance (MANOVA). The two-way MANOVA determined the experimental effects of message framing and information source on the dependent variables (information recall, trust, source expertise, source credibility, and anticipated consumption behavior), concurrently [48]. We then used a Discriminant Function Analysis (DFA) to follow up a significant MANOVA. It is recommended to use a DFA to follow up a significant MANOVA because it is “closely aligned to the study of effect determined by a multivariate analysis of variance” [49] (p. 30). DFA is used to determine which outcome variables contributed the most to separating independent variables [50,51,52].

## 3. Results

The means and standard deviations of the dependent variables for each of the treatment groups are presented in Table 2. The meat scientist delivering the narrative frame message had the highest mean for information recall (M = 2.52, SD = 1.16) with the consumer delivering the analytical frame message having the lowest mean for information recall (M = 2.13, SD = 1.38). The meat scientist delivering the narrative frame message had the highest mean for trust (M = 5.96, SD = 1.01) with the consumer delivering the analytical frame message having the lowest mean for trust (M = 5.63, SD = 0.91). The meat scientist delivering the narrative frame message had the highest mean for source expertise (M = 6.13, SD = 0.93) with the consumer delivering the narrative frame message having the lowest mean for source expertise (M = 5.24, SD = 1.43). The meat scientist delivering the narrative frame message had the highest mean for source credibility (M = 6.10, SD = 0.87) with the reporter delivering the analytical frame message having the lowest mean for source credibility (M = 5.59, SD = 1.01). Lastly, the producer delivering the narrative frame message had the highest mean for anticipated consumption behavior (M = 4.60, SD = 0.89) with the reporter delivering the narrative frame message having the lowest mean for anticipated consumption behavior (M = 4.28, SD = 0.78).

Our first research question looked at whether there was a significant interaction between message frame and information source on the combined dependent variable set (information recall, trust, source expertise, source credibility, and anticipated consumption behavior). A statistically significant effect was not obtained (*p* = 0.61), Pillai’s Trace = 0.05, F (15, 701.58) = 0.86.

When looking specifically at message frame, the information recall mean was higher for the analytical frame (*M* = 2.35, *SD* = 1.18) than it was for the narrative frame (*M* = 2.33, *SD* = 1.20). The trust mean was higher for the narrative frame (*M* = 5.85, *SD* = 1.05) than it was for the analytical frame (*M* = 5.75, *SD* = 0.96). The source credibility mean was higher for the analytical frame (*M* = 5.82, *SD* = 1.01) than it was for the narrative frame (*M* = 5.95, *SD* = 0.98). Finally, the anticipated consumption behavior mean was higher for the narrative frame (*M* = 4.44, *SD =* 0.91) than it was for the analytical frame (*M* = 4.39, *SD* = 0.88; Table 3).

Research question two sought to determine if there are significant differences in the combined dependent variables (information recall, trust, source expertise, source credibility, and anticipated consumption behavior) for message frame (analytical/narrative). A MANOVA for the main effect for message frame was not statistically significant (*p* = 0.54), Pillai’s Trace = 0.02, *F* (5, 254) = 0.81, indicating no significant differences in the combined dependent variable set for message frame.

As reported in Table 4, the meat scientist information source had the highest mean score for information recall (M = 2.43, SD = 1.08), trust (M = 5.89, SD = 1.03), source expertise (M = 6.08, SD = 0.96), and source credibility (M = 6.03, SD = 0.97) with the producer information source having the highest mean score for anticipated consumption behavior (M = 4.46, SD = 0.96). In contrast, the consumer information source had the lowest mean for information recall (M = 2.21, SD = 1.29), trust (M = 5.69, SD = 1.08), and source expertise (M = 5.30, SD = 1.31) with the reporter information source having the lowest mean for source credibility (M = 5.61, SD = 1.06) and anticipated consumption behavior (M = 4.38, SD = 0.77).

Research question three sought to determine if there are significant differences in the combined dependent variables of information recall, trust, source expertise, source credibility, and anticipated consumption behavior for information source (i.e., consumer, producer, reporter, and meat scientist). A statistically significant MANOVA effect was obtained (*p* < 0.001), Pillai’s Trace = 0.15, F (15, 701.58) = 2.67. The partial effect size was 0.05 (η^2^ = 0.05), indicating a small effect [53]. Discriminate function analysis (DFA) was used to determine which weightings of the dependent variables best discriminated between the four sources of information. As shown in Table 5, DFA revealed functions 1 through 3 as significant, Wilk’s lambda λ = 0.86, χ^2^ (15) = 39.45, *p* < 0.001, Rc^2^ = 0.10. Therefore, the discriminant function explained 10.24% of the variance among information source (consumer, producer, reporter, and meat scientist).

Both an inspection of the structure matrix and canonical coefficient table confirmed the importance of source expertise. Source expertise had the largest value for the structure matrix (0.79). Variables with a structure matrix value larger than 0.3 are considered most meaningful in the discriminant function [54]. Based on this, source expertise is the only discriminating variable considered in the discriminant function. Additionally, source expertise had the largest canonical coefficient (−1.42). Variables with standardized canonical coefficients larger than the absolute value of half of the largest standardized canonical coefficient can be included in the discriminant function [54]. Therefore, we established the cut-off value at 0.71 (−1.42/2 = |−0.71|), and source expertise was the only identifiable discriminating variable for information source.

## 4. Discussion

Based on our review of the literature, our study is the first to empirically examine how message frame and information source impact consumers’ attitudes toward the AAACS. Thus, though we recognize there is more work to be done in this domain, we believe that our study provides foundational evidence about novel food technology messaging and consumers’ preferred information sources, especially in the study of the AAACS.

First, we did not find a significant interaction effect between message frame and information source, indicating that these variables are independent of one another. Thus, the information source and message frame will independently impact information processing when communicating about the alternative meat curing system studied herein.

Second, we did not detect a significant difference between the narrative and analytical message frame on the combined dependent variable set. Similar results exist in the study of the nonsignificant effect of framing on consumers’ behavior toward GM food. For example, previous research has indicated that the type of frame did not significantly influence consumers’ actual behavior change toward innovative food products [55]. Conversely, other studies identified significant effects of framing on consumers’ acceptance of meat innovations. For example, it has been found that different framing strategies had a significant effect on consumers’ behavioral intentions toward cultured meat [56]. Additionally, narrative-framed information has been found to reduce consumers’ negative perceptions toward novel food technologies [33]. Our study’s results may be inconsistent with previous framing research because the studies cited above did not look at a combined set of dependent variables as was examined in our study. It is likely that the effects of message framing may be too complex to be measured by single outcome variables. Because of this, the results of our study likely provide depth to the literature. Message framing studies conducted in the past have looked at only one outcome variable or examined multiple outcome variables independently. These studies have not looked at the interaction between multiple independent variables on a combined dependent variable set. Since the heterogeneous findings regarding the effects of framing on novel food product communications, future research should further investigate other framing strategies beyond conventional narrative and analytical framing. Different strategies could increase public understanding and engagement with novel food technologies. Additionally, other framing strategies could be investigated in other dissemination mediums besides video (e.g., textual messages, interactive messages, and audio messages).

Third, significant differences between the combined dependent variables of information recall, trust, source expertise, source credibility, and anticipated consumption behavior for information source (producer, consumer, reporter, or meat scientist) was found. Specifically, there were higher levels of information recall, greater levels of trust, and higher levels of source expertise and source credibility when messages were delivered by the meat scientist compared to messages delivered by the consumer, producer, or reporter. These findings are consistent with previous studies that indicate people listen more closely to individuals who are deemed experts [57]. Thus, our study provides empirical evidence to inform those who communicate about novel food technologies, especially those communicating about different types of meat curing systems. Based on our study, we recommend that the primary information source be the AAACS content expert. This does not negate the importance of applying effective communication practices and techniques within the content area. In fact, applying such practices is as important as the content itself—one cannot be sacrificed for the other. Thus, it is imperative that agricultural communication practitioners form a partnership with the content expert to deliver a message that consumers will use to make informed decisions about their food consumption. This partnership could look differently, depending on the context and the content. For example, the practitioner could teach scientists and agriculturalists effective strategies to better connect with diverse audiences when communicating about their work. In summary, based on the three defined research questions, our study provides three primary contributions: (a) no significant interaction between message frame and information source was found, (b) a significant difference for the combined dependent variable of information recall, trust, source expertise, source credibility, and anticipated consumption behavior for message frame was not found, and (c) a significant difference for the combined dependent variable was found for information source.

We recognize our study is not without limitations. First, we did not consider the participants’ ability to evaluate analytical and narrative information. Consumers’ skeptical attitude toward scientific information about meat-based products could be explained by their insufficient ability to evaluate scientific messages [58]. The dependent variable of information recall was included in part to provide insight into how well participants were able to evaluate the narrative and analytical information. This data helped to better understand if participants were able to comprehend the key information within the video. It is recommended that further data analysis take place to better understand how the successful evaluation of information impacts other variables such as trust and anticipated behavior. Additionally, future studies should include more measures to better understand participants’ ability to evaluate information.

Second, although our study includes a diverse sample, all participants were students at one university. Thus, caution is recommended in generalizing our results. However, it is common knowledge that university graduates have higher earning power and become powerful contributors to gross domestic product. This initial study provides important foundational evidence addressing the research questions. With that, our study should be replicated using a national sample of consumers. Replications of this study can further evaluate unexpected results and collect further information to confirm and draw more meaningful conclusions. Additionally, complementary research on the information influence mechanism is recommended to draw valuable conclusions.

## 5. Conclusions

Consumers might be skeptical about novel agri-food technologies because of unfamiliarity and uncertainty. As a result, negative attitudes influence the adoption of such technologies. Our study evaluated if the message frame and/or information source influence individuals’ information recall, trust, source expertise, source credibility, and anticipated consumption behavior of meat products cured using the amino acid-based alternative meat curing system (AAACS). We found that when a source directly involved with the AAACS innovation communicated about the technology, this consumer group had greater information recall, trust, source expertise, source credibility, and anticipated consumption behavior. Particularly, participants perceived a higher level of source expertise for the meat scientist compared to the source expertise for the reporter and the consumer. Additionally, participants indicated higher levels of information recall, trust, source expertise, and source credibility from messages delivered by the meat scientist and participants rated higher levels of anticipated consumption behavior when messages were delivered by the producer. Our study contributes to the field of communication efforts regarding meat products produced with innovative technologies. As products cured with the AAACS are introduced into the market, the most effective approach to promoting the product is through the meat scientist or content expert. Their expertise can be used to develop effective communication messages that increase public understanding and build trust between consumers and scientists. We recommend that communication practitioners who are preparing information about the AAACS prioritize strategically selecting information sources directly involved with the technology. Additionally, agricultural and science communicators should focus on educating meat scientists about effective communication strategies and work with experts to combine expertise–communications and meat science–to deliver an effective message about meats cured using the AAACS.

## Figures and Tables

**Table 1 foods-12-01535-t001:** Confirmatory Factor Analysis Results of Factor Loading and Reliability.

Variable	Item	Standardized Factor Loadings	Cronbach’s Alpha
Trust	Trustworthiness	0.89	0.90
	Sincerity	0.88	
	Reliability	0.80	
	Dependability	0.78	
	Honesty	0.70	
Source Expertise	Experience	0.90	0.94
	Knowledge	0.90	
	Expertise	0.85	
	Qualified	0.84	
	Skill	0.83	
	Competence	0.77	
Source Credibility	Goodwill	0.94	0.87
	Openness	0.84	
	Integrity	0.70	

Note. Each item was measured on a seven-point semantic differential scale, using bipolar adjectives (1 = negative; 7 = positive).

**Table 2 foods-12-01535-t002:** Mean Comparisons Between Message Frame and Information Source by Information Recall, Trust, source Expertise, Source Credibility, and Anticipated Consumption Behavior.

Test		Information Recall ^a^		Trust ^b^		Source Expertise ^b^		Source Credibility ^b^		Anticipated Consumption Behavior ^c^	
	*n*	*M*	*SD*	*M*	*SD*	*M*	*SD*	*M*	*SD*	*M*	*SD*
Consumer											
Analytical	32	2.13	1.38	5.63	0.91	5.36	1.19	5.95	0.97	4.28	0.70
Narrative	35	2.28	1.22	5.74	1.23	5.24	1.43	5.94	0.98	4.53	0.87
Producer											
Analytical	32	2.44	1.16	5.92	0.91	5.91	1.07	5.81	1.02	4.31	1.01
Narrative	34	2.38	1.29	5.82	1.02	5.65	1.00	6.10	0.87	4.60	0.89
Reporter											
Analytical	34	2.48	1.19	5.66	0.98	5.41	1.11	5.59	1.01	4.49	0.74
Narrative	34	2.17	1.15	5.88	0.91	5.50	1.11	5.63	1.13	4.28	0.78
Meat Scientist											
Analytical	33	2.35	1.00	5.81	1.03	6.04	1.00	5.92	1.05	4.47	1.03
Narrative	32	2.52	1.16	5.96	1.01	6.13	0.93	6.15	0.88	4.34	1.06

Note. *M* denotes mean, and *SD* denotes standard deviation. ^a^ Score ranging from 0–4, based on the number of multiple-choice questions answered correctly. ^b^ Sets of bipolar adjectives measured on seven-point semantic differential scales (1 = negative; 7 = positive). ^c^ Items measured with six-point Likert scales (1 = strongly disagree; 6 = strongly agree).

**Table 3 foods-12-01535-t003:** Means Comparisons for Message Frame by Information Recall, Trust, Source Expertise, Source Credibility, and Anticipated Consumption Behavior (n = 266).

Dependent Variable	Analytical (*n* = 131)		Narrative (*n* = 135)	
	*M*	*SD*	*M*	*SD*
Information Recall ^a^	2.35	1.18	2.33	1.20
Trust ^b^	5.75	0.96	5.85	1.05
Source Expertise ^b^	5.68	1.12	5.62	1.17
Source Credibility ^b^	5.82	1.01	5.95	0.98
Anticipated Consumption Behavior ^c^	4.39	0.88	4.44	0.91

Note. *M* denotes mean and *SD* denotes standard deviation. ^a^ Score ranging from 0–4, based on the number of multiple-choice questions answered correctly. ^b^ Items measured with seven-point semantic differential scales, using bipolar adjectives (1 = negative; 7 = positive). ^c^ Items measured with six-point Likert scales (1 = strongly disagree; 6 = strongly agree).

**Table 4 foods-12-01535-t004:** Means Comparisons for Information Source by Information Recall, Trust, Source Expertise, Source Credibility, and Anticipated Consumption Behavior.

Dependent Variable	Consumer (*n* = 67)		Producer (*n* = 66)		Reporter (*n* = 68)		Meat Scientist (*n* = 65)	
	*M*	*SD*	*M*	*SD*	*M*	*SD*	*M*	*SD*
Information Recall ^a^	2.21	1.29	2.41	1.22	2.33	1.17	2.43	1.08
Trust ^b^	5.69	1.08	5.87	0.97	5.77	0.95	5.89	1.03
Source Expertise ^b^	5.30	1.13	5.78	1.04	5.45	1.10	6.08	0.96
Source Credibility ^b^	5.95	0.97	5.96	0.95	5.61	1.06	6.03	0.97
Anticipated Consumption Behavior ^c^	4.41	0.80	4.46	0.96	4.38	0.77	4.41	1.04

Note. *M* denotes mean. *SD* denotes standard deviation. ^a^ Score ranging from 0–4, based on the number of multiple-choice questions answered correctly. ^b^ Items measured with seven-point semantic differential scales, using bipolar adjectives and phrases (1 = negative; 7 = positive). ^c^ Items measured with six-point Likert scales (1 = strongly disagree; 6 = strongly agree).

**Table 5 foods-12-01535-t005:** DFA for Combined Dependent Variables Based on Information Source.

Variables						Structure Matrix	Standardized Canonical Coefficient
Information Recall						0.21	0.31
Trust						0.22	−0.51
Source Expertise						0.79	−1.42
Source Credibility						0.21	0.33
Anticipated Consumption Behavior						0.01	0.31
Function	λ	Χ^2^	*df*	*p*	Eigenvalue	Canonical Correlation	
1 through 3	0.86	39.45	15	<0.0001	0.12	0.32	
2 through 3	0.96	10.61	8	0.23	0.04	0.20	
3	1.00	0.37	3	0.95	0.00	0.04	

Note. R_c_^2^ = 0.10.

## Data Availability

https://doi.org/10.18738/T8/D9WLGB.

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
