# Peer review of "Effects of Message Framing and Information Source on Consumers’ Attitudes toward an Amino Acid-Based Alternative Meat Curing System"

_foods, 2023, doi:10.3390/foods12071535_

Round 1

Reviewer 1 Report

This topic is interesting and the design is well, but there are some points need to be revised. The comments on the content are as following:

1. The findings of this study suggest that effective communication about food technologies like the AAACS is critical in order to gain consumer acceptance. Message frame and information source are two important peripheral cues which can have a significant impact on attitudes towards novel foods, with source expertise being the most powerful discriminating variable for assessing attitude. This has implications for marketers looking at ways they can effectively communicate new products or services which may be met by skepticism due their novelty factor among consumers. It suggests that emphasizing expert opinion could help build more positive perceptions around such innovations as well as increase acceptance levels amongst target audiences through increased knowledge sharing regarding its benefits over existing alternatives available in market today .

2. The data is not representative and cannot support the author's research. The research group on meat products cured using the AAACS can not be limited to college students at one university.  In lines 157-158, it should also be explained why the author selected the number of questionnaires from 470 to 266. The age, education level, income level and other characteristics of participants should be supplemented. The results obtained for such samples are biased.

3. In lines 166 through 188, the author did not make clear how narrative messages and analytical messages were presented in the study, . And consumers themselves have different ways of thinking, so different consumers react differently to narrative messages and analytical messages. This is not taken into account in this study, so the model is endogenous.

4. In the “2.2.3. Dependent Variables” section, it may not be representative and reasonable to use only information recall as a variable to measure the attitudinal cognitive component. Moreover, it would be better if you could use the literature to show why Trust, source expertise, and source credibility were chosen to measure the affective component of an attitude. This may be the reason why no significant interaction effect between message frame and information source, and no significant difference between the narrative and analytical message frame on the combined dependent variable set. Complementary research on the information influence mechanism is needed to draw valuable conclusions.

5. The study design is innovative and the main problem is the value and credibility of the results. Only one of the three questions that the article sought to verify was answered, after that, only 10.24% of the variance was explained in the Discriminant Function Analysis for the only significant difference. As expressed in the article, the results of the article (i.e., no significant differences) are also inconsistent with the results of some previous articles. In addition to the limitations of the sample range mentioned in the article that warrant expansion of the study, a large reason for the insignificant results to the extent that the conclusions are inconsistent with previous studies may be the sample size issue, 266 valid samples were divided into 8 groups for the experiment, so the study does deserve more exploration. However, it is undeniable that the authors found a very interesting problem and designed a very clever experiment to explore this problem, the conclusion and suggestions from the study do contribute to solving real-world problems.

Author Response

Reviewer 1:

  • Noted that the data is not representative and cannot support the research.
    • Authors’ Comments/Responses: We referring to the sample, “participants” was replaced in most instances to “students.” This way, it was very transparent that the sample was made up of students from a university.
  • Suggested that the age, education level, income level, and other characteristics of participants be supplemented.
    • Authors’ Comments/Responses: To address this, more demographic information about the sample was included in lines 173–185 This allowed readers to better see the variation in participants gender, ethnicity, country of origin, university college, education, and average meat consumption. Additionally, a recommendation was included to replicate the study using a larger, national sample in lines 504–508.
  • Recommended that we explain why the selected number of questionnaires went from 470 to 266.
    • Authors’ Comments/Responses: To address this, an explanation of cleaning the data was included in lines 166–167.
  • Suggested that the author did not make clear how narrative messages and analytical messages were presented in the study.
    • Authors’ Comments/Responses: To address this comment, we elaborated on how narrative and analytical messages were presented in the study in section 2.2.1. Additionally, we have included the supplemental materials of video scripts to provide more clarity about the narrative and analytical message scripts. Furthermore, we are including a new story about the AAACS as supplementary material, as well.
  • Stated that it may not be representative and reasonable to use only information recall as a variable to measure the cognitive component of an attitude.
    • Authors’ Comments/Responses: We reword the way that the dependent variables were framed in the context of measuring the tripartite classification of attitudes. The paragraph including lines 244 to 254 was reworded so that it did not sound like information recall alone measured. Additionally, the paragraph was reworded so it does not sound like three dependent variables measured the entire affective component of an attitude and that anticipated consumption behavior was not a complete measure of the behavioral component of an attitude. There are endless ways to measure each component of an attitude (behavioral, cognitive, and affective or attitudinal). This is why we chose to rephrase the paragraph to demonstrate that each component of an attitude was represented by at least one variable, but variables did not measure each component in full. Additionally, we explained why information recall was chosen as a measure for the cognitive component of an attitude. We also elaborated on how the dependent variable was designed and why it was designed in that manner (see lines 234-296). In the discussion section of the article, in lines 490–499, we recommended future studies include additional cognitive measures (e.g., intermediate and long-term recall), as well.
  • Suggested using the literature to show why trust, source expertise, and source credibility were chosen to measure the affective component of an attitude.
    • Authors’ Comments/Responses: We elaborated on why the dependent variables and items within those variables were selected. We included literature to support the dependent variables. Additionally, we added more information about the instrument development process to further clarify how the dependent variables were selected and developed.
  • Noted that complementary research on the information influence mechanism is needed to draw valuable conclusions.
    • Authors’ Comments/Responses: This was included as a recommendation in lines 505–509.
  • Noted a potential problem with the value and credibility of the results of this study.
    • Authors’ Comments/Responses: To address this, a recommendation was included to continue research in this area with a larger, national sample. It was also recommended that more data be collected to further evaluate unexpected results and confirm any conclusions drawn from this study.

Reviewer 2 Report

The manuscript addresses a very important issue, namely how to communicate information to consumers so that consumers gain knowledge about food (here, new technology), form attitudes towards it and make the best choices for themselves.

A strength of this study is the use of two theoretical frameworks, which makes it possible to see what new contributions the study makes to existing knowledge. The empirical study was prepared and conducted with great care. It is a pity that a video description, including a transcript of the information provided, was not included as a supplement. The information provided in the article on the content presented in the video is quite general.

The study has its limitations, but the authors have written about them. The biggest limitation is that participants' ability to evaluate analytical and narrative information was not taken into account.

My criticism relates primarily to the first paragraph, especially lines 34-35. This is one of many recommendations relating to meat consumption. Most of them encourage a reduction in meat consumption for very different, well-known reasons. The statement is bold and does not take into account those people who consume a great deal of meat and meat products.  In my opinion, the rationale for changes in technology is to improve the quality of the food, even if it will be consumed in smaller quantities, and this is how I would run the rationale for this technological change.

Author Response

Reviewer 2:

  • Suggested that a transcript of the videos be included as a supplement because the information provided in the article about the content presented in the video is general.
    • Authors’ Comments/Responses: To address this, more information was included about how each message script was created and presented. Additionally, we are including the message scripts as supplementary material.
  • Mentioned that the biggest limitation is that participants’ ability to evaluate analytical and narrative information was not considered.
    • Authors’ Comments/Responses: This limitation was included at the end of the discussion section. We expanded on ways that we had addressed this limitation in the study and included recommendations to address this limitation.
  • Noted that lines 34-35 regarding meat consumption were bold and do not take into account those people who consume a great deal of meat and meat products.
    • Authors’ Comments/Responses: To address this comment, paragraph 1 (lines 28-38) was deleted. After review, we felt the entire paragraph was not directly pertinent to research questions addressed in this article. Beginning with the following paragraph provided a far more direct way to introduce the article.
  • Recommended running the rationale for technological change to improve the quality of food, even if it is consumed in smaller quantities.
    • Authors’ Comments/Responses: Paragraph 1 was deleted. Additionally, information was deleted in lines 56-57 that mentioned guidelines consumers have received to reduce meat consumption. This way, the purpose of the AAACS is more focused on the demand for alternative meat curing methods.

Reviewer 3 Report

Manuscript ID: foods-2244574

Title: Effects of Message Framing and Information Source on Consumers’ Attitudes Toward an Amino Acid-Based Alternative Meat Curing System

The manuscript is very interesting, well written and comprehensive. The communication with consumers about new methods used in meat production is very important. The authors studied the effects of various factors facing the consumer acceptance of innovations in meat technology.

I have some suggestions for the Authors:

1) L233. Authors state that they used a “six-point Likert scale of 1 = strongly disagree, 2 = disagree, 3 = somewhat disagree, 4 = somewhat agree, 5 = agree, 6 = strongly agree “. Please explain to the readers why you decided to cut off the middle value: neither agree, nor disagree.  

2) Table 2, 3, 4 and L206. Please check again the scales used for information recall, because it seems to me that there is a typo in the text. In L206 it is mentioned a 0-4 scale, while in the footnote of the tables (a), it is written a 1-4 scale.

3) Table 2, 3, 4 Authors state: aScore ranging from 1–4, based on the number of multiple-choice questions answered correctly. Please revise the statement. Do you mean “correctly” or “fully”? You already mentioned in paragraph 2.1 that from 470 participant remained only 266 after cleaning the data.

Author Response

Reviewer 3:

  • Asked for an explanation of why no middle value (neither agree, nor disagree) was included in the Likert scales.
    • An explanation was included in lines 321–327 of the article.
  • Recommended checking the scales used for information recall, as there appears to be a typo in the text. Line 206 mentions a 0–4, while the footnote mentioned a 1–4 scale.
    • Each footnote mentioning a 1–4 scale was changed to a 0–4 scale. This was a typo. Because participants could potentially answer no multiple-choice questions correctly, an information recall score could range from 0 to 4.
  • Recommended revising the statement “Score ranging from 1–4, based on the number of multiple-choice questions answered correctly.” The reviewer asked for clarification whether the authors meant correctly or fully, based on a reference to the way data was cleaned.
    • A typo in the footnotes about how the information recall scales are measured likely caused confusion regarding exactly what the scales are measuring. Each of the footnotes were edited to read a score ranging from 0–4. The scales measure the number of correctly answered information recall questions, which could be any score from 0 to 4. Additionally, more information was added regarding how the data was cleaned, so there was not confusion about data being cleaned and information recall scores.